# Development and Usability Validation of a Social Robot Platform for Physical and Cognitive Stimulation in Elder Care Facilities

**DOI:** 10.3390/healthcare9081067

**Published:** 2021-08-19

**Authors:** Luis Cobo Hurtado, Pablo Francisco Viñas, Eduardo Zalama, Jaime Gómez-García-Bermejo, José María Delgado, Beatriz Vielba García

**Affiliations:** 1CARTIF Technology Centre, 47151 Valladolid, Spain; pabvin@cartif.es (P.F.V.); ezalama@eii.uva.es (E.Z.); jaigom@eii.uva.es (J.G.-G.-B.); 2ITAP-DISA, Department of Systems Engineering and Automation, School of Industrial Engineering, University of Valladolid, 47002 Valladolid, Spain; 3LACORT, Lacort Elderly House, 47150 Valladolid, Spain; josemaria@residenciaslacort.com (J.M.D.); bea@residenciaslacort.com (B.V.G.)

**Keywords:** robotics, social, robots, human/machine interaction, robot programming, physical stimulation, cognitive stimulation, elder care

## Abstract

This article shows our work for developing an elder care platform for social interaction and physical and cognitive stimulation using the Pepper robot and Android OS as clients, based on the knowledge acquired on our long-term social robotics research experience. The first results of the user’s acceptance of the solution are presented in this article. The platform is able to provide different services to the user, such as information, news, games, exercises or music. The games, which have a bi-modal way of interacting (speech and a touch screen interface), have been designed for cognitive stimulation based on the items of the mini-mental state examination. The results of the user’s performance are stored in a cloud database and can be reviewed by therapists through a web interface that also allows them to establish customized therapy plans for each user. The platform has been tested and validated, first using adult people and then deployed to an elder care facility where the robot has been interacting with users for a long period of time. The results and feedback received have shown that the robot can help to keep the users physically and mentally active as well as establish an emotional link between the user and the robot.

## 1. Introduction

As people age, some physical and cognitive disabilities or social isolation situations might appear that affect their ability to perform activities in a self-sufficient manner. The use of social robots to assist these people can help them to maintain or improve their capacities through social interaction and the realization of different activities to work on their mental and physical abilities, contributing to the improvement of their quality of life.

Social robots are robots that are able to interact and communicate with people using language, behavior, patterns and social standards in different environments such as schools, hospitals, workplaces or homes.

The ageing population, especially in western countries, makes it necessary to improve elder care dedication. Due to the rising number of old people (it is estimated that by 2050, the population over than 80 years of age will have increased very significantly) [1], the current number of people specialized in elder care will become insufficient. This requires the emergence of new technologies that can help in the care field, facilitating the independent life of the elderly. Social robotics could be an important factor through the use of robots that are able to interact with those people, contributing to the avoidance of social isolation, physical inactivity and sedentary lifestyles.

Currently, there are different social robots already available as commercial products or prototypes such as Nao, Pepper (Softbank Robotics), Asimo [2] or iCub [3].

On the other hand, service robots are those who help elderly users to maintain an autonomous lifestyle, helping to perform daily tasks such as eating, dressing or going to the bathroom, mobility and displacements, house maintenance or monitoring of certain tasks where continuous surveillance is needed. There are a few robots of this kind such as the nurse robot Pearl [4], the small pet robot iCat [5], the assistant Care-o-bot [6] or the Robocare project [7].

There are also other kind of robots that instead of providing certain services, focus on serving as a companion to the person, trying to establish an emotional link with them that could improve their mood, entertain them or decrease their feelings of loneliness. Some robots of this kind are the seal robot Paro [8], the teddy bear Huggable [9] or the dog robot Aibo from Sony [10].

The convenience of using a social robot instead of a tablet-only interface has been addressed in several studies, such as learning contexts for children [11,12], healthcare delivering instructions [13] or smart environments [14,15,16], showing that people respond better to social robots than a tablet or touch screen only interface. Other studies have also used socially assistive robots to perform cognitive therapies for people with dementia achieving positive results [17]. Physical rehabilitation using social robots has also been addressed with promising results on elderly users [18,19]. Those and other studies [20] show that the use of social robots for elderly care is a field of research with great potential.

## 2. Materials and Methods

### 2.1. Description of the Robot Platform and Its Limitations

Our research aims to develop and validate a social robot platform using Pepper robot and Android OS as clients that allows it to serve as an assistant robot for elder care, making it able to interact socially with them, entertain them and keep them physically and mentally active. The first results of the user’s acceptance of the solution are presented in this article.

Pepper is a 120-centimeter-tall humanoid robot developed by Softbank Robotics [21]. It is designed to interact with people and can be programmed and controlled using custom Android apps.

In order to achieve success in our goal, there are some limitations that must be considered to be able to overcome them. In the first place, there are current limitations of social robots and the ones specific to Pepper robot. These include the following:Difficulties to have a fully reliable facial recognition.Scripted dialogues (the robot has no real “intelligence”): although deep learning natural language understanding is used to recognize the user’s intent, those intents and robot responses have to be programmed manually.Limited Software Development Kit (SDK): as the Pepper robot is programmed through the Android tablet, developers cannot have direct access to robot sensors or actuators, only to the functions and methods implemented by Softbank Robotics in their android SDK.

In the second place, the following peculiarities associated with interaction with elderly people must also be taken into account:Visual impairments: the user interface will have to have large buttons and letters to ensure that they are more easily readable.Hearing loss: the robot will need to speak clearly and at an appropriate volume and pitch.Digital divide: the interface and interaction with the robot must be simple, natural and intuitive to ensure that older people can interact with it without problems.

### 2.2. Cognitive State Assessment in the Elderly by Using the Mini-Mental State Examination

A comprehensive psychiatric evaluation in the elderly includes a comprehensive mental status examination that involves integrating components of both medical and psychiatric clinical models. The classic medical model pursues symptoms and signs in a problem orientated fashion, seeking to match them with a single unifying diagnosis and then designing an appropriate treatment strategy leading to a cure. This approach is less useful when dealing with geriatric patients. With these patients, aspects of the psychiatric clinical model are emphasized, i.e., ongoing treatment rather than cure, and chronic rather than acute care given by an interdisciplinary team.

The strategy in the geriatric care is focused on maximizing and maintaining individual functions and behavior, giving the best quality of life to the elderly. It is less involved in the disease and more with disability.

Richards and Maletta in [22] highlight the importance of performing a standard, valid and reliable cognitive examination to achieve an acceptably comprehensive Mental State Examination (MSE). In this regard, they recommend the Mini-Mental State examination (MMS) as a good example of an MSE test that can be performed.

The MMS is a test focused on the cognitive aspects of mental functions, and excludes questions concerning mood, abnormal mental experiences and the form of thinking [23].

The MMS’s questionnaire should be asked in the order listed and scored immediately. The role of the tester (psychiatric resident, nurse or volunteer) is very important because it is needed to create a comfortable ambience to the patient during the test, avoiding a pressure on items that the patient finds difficult. In this setting, most patients cooperate. In our research, we are using the MMS version that has a maximum score of 35. The test is divided into two parts; the first of which requires vocal responses only and covers orientation, memory and attention, and has a maximum score of 24. The second part tests the ability to name, follow verbal and written commands, write a sentence spontaneously and copy a complex polygon similar to a Bender–Gestalt, with a maximum score of 11. As this part involves reading and writing, the users with severely impaired vision may have some extra difficulty. The professional might have to apply some solutions, such as larger figures or bigger font sizes. Those adaptations are allowed to be performed during the evaluation of the user. [24,25].

### 2.3. System Architecture and Features

The system has been developed with the aim of providing a range of functionalities for the robot that can be useful to interact with the user. In this case, the target users are elderly people; therefore, the functionalities are focused on this type of user, seeking to maintain and stimulate their physical and mental activity while entertaining them. An overview of the system architecture is shown in Figure 1.

The interface to the system is an Android App that uses the Pepper Android SDK to communicate with the robot and send orders such as sending the phrases to say or the animations to play. It has different views to cover all the features offered, such as a user selection screen, information, games, exercises or music. It also has support for multiple languages, specifically Spanish and English for the moment.

The Android App connects with a web service to exchange information such as user data and statistics that are stored on a database. It also communicates with Dialogflow in order to perform natural language understanding (NLU) from the user’s speech and handle the conversation flow. Other web APIs such as Wikipedia or OpenWeatherMap are also used to receive specific information. It offers a bimodal interface, allowing the user to navigate through the menus or request services either by using the touch screen or by speech using NLU.

The App is opened automatically when turning on the robot, showing the user selection menu where the user can select their profile by touching on the screen or saying their name. New users can also be created and allows the user to take a photo of themself that will be shown on the user selection menu.

After the user is selected, the proper interaction is started as the robot greets the user. Then, the main menu is shown on the screen, showing the following main categories of the different services available: information, games, exercises and music (Figure 2).

An overview of the services offered by the robot is shown in Figure 3. The offer of services has been chosen to focus on the needs of elderly users; therefore, it has been agreed with the therapists based on their experience and work with them. As one robot is kept permanently in the elder care facility interacting with the users, improvements and new features are continuously being added based on the feedback received using a remote update system integrated in the App, in a continuous process of co-creation.

All the user’s personal data are stored in the database through the web service, which handles all data input and output. This allows the robot to speak with the person and remember some personal data such as their age, birthdate, birthplace or current living place.

The interface of the menus has been designed to be simple and shows big buttons and big font sizes in order to facilitate the readability and interaction for elderly people.

#### 2.3.1. Information

The services to obtain information include meteorology, news, information search and curiosities.

Regarding the meteorology, the OpenWeatherMap web site has been selected, since it allows the user to obtain the current climate conditions through its API to ensure that the robot can say them and show the corresponding data on the screen.

Regarding the news, the robot has to be able to obtain the news from the Internet and tell them. To do so, it uses RSS feeds from different newspapers to provide front page, economic, sports and local news. The rss2json web API has also been used to obtain the content in a JSON format [26].

The information search function consists of the robot being able to explain or talk about some topics, be it a city, country, historical figure, sport, etc. The source chosen to obtain the information has been Wikipedia, since it is a free source that also offers an API that allows the content of the articles to be obtained directly in a JSON format.

In addition, data are also obtained from the database through the web service, where different customized terms are collected with their corresponding explanation. This allows the robot to talk about data customized for the user, such as their birthplace or hobbies.

#### 2.3.2. Games

The games that the robot offers have been designed for entertainment and cognitive stimulation aligned to the axes of the MMS examination. Thus, the developed games are focused on the following four different categories: orientation, calculation, memory and language. The bi-modal user interface allows them to work using the speech and/or the touch screen interface.

In Figure 4, examples of the orientation game “flags” and the memory game “pairs” are shown.

In Figure 5, examples are also shown of the calculation game “addition and subtraction” and the language game “sayings”.

The games feature different levels of difficulty that make them easier or harder in order to adapt to the cognitive abilities of the user and their evolution over time. All the result scores are stored in the cloud database, allowing their progression to be tracked and studied over time.

The system also allows the therapists to set up customized activity plans for each user or group of users through a web interface. Those plans can be then executed by the robot, allowing work on the relevant categories according to the cognitive state of each user.

#### 2.3.3. Exercises

A series of physical exercises have been designed for the elderly and are divided into neck exercises, hand exercises, shoulder or arm exercises and various exercises. To keep the user active and motivated, each group of exercises is performed with background music, the robot explains each exercise aloud and lists each repetition. An example of the robot performing shoulder exercises can be seen in Figure 6.

#### 2.3.4. Music

The robot can play music when the user requests it, through voice or the Tablet menus, offering different styles adapted to the tastes of the elderly: classical, popular, modern or copla.

To display the desired music on the screen, customized YouTube playlists with songs of the requested genre are played. These lists have been created specifically for the robot using its own account, which on the one hand, ensures that the videos are played without problems (some do not allow them to be embedded), and on the other hand, that these music lists can be remotely modified.

## 3. Results

### 3.1. Description of the Tests Conducted

To evaluate the performance of the robot, a series of tests have been carried out in two phases, first with adult volunteers and then in an elder care facility. In both cases, they interacted with the robot in their native language (Spanish).

The first phase was carried out with the staff of the Cartif technology center. A total of 38 participants with an average age of 38 years old interacted individually with the robot. Before starting the experiment, each participant was given brief instructions, explaining that they could interact with the robot by talking or touching the screen and if they did not know what to do, they could ask “what can you do?” and then the robot would tell them the different services that it can provide.

Each of the participants interacted freely with the robot for ten minutes while being observed from a different room. At the end of the interaction, they were given a questionnaire to complete. The questionnaire used is based on the Almere model, designed to evaluate the acceptance of people towards a social assistive robot [27], to which an assessment of the different functionalities of the robot has been added. The questionnaire contains 33 questions that are answered using a 5-points Likert scale from one to five (totally disagree—disagree—do not know—agree—totally agree) and are used to evaluate different aspects of the interaction. Additionally, we also included nine questions to evaluate the different robot features from one to five (one, lowest rating; five, highest rating) in order to find drawbacks and which of them they appreciated the most.

The second phase of the tests consisted of testing the performance of the robot in the operational environment of a nursing home. In this case, we have had the collaboration of the Lacort Viana elder care facility, located in the town of Viana de Cega, in the province of Valladolid (Spain). The robot was moved to the center and has stayed there for a period of over a year, being located in an office where the caregivers have been taking different residents to interact with it, and also using it for group interactions in bigger rooms, as can be seen in Figure 7. A total of 21 participants were chosen to interact with the robot, with an average age of 88 years old and an average MMS score of 23 (35 being the maximum score that can be achieved). The results shown were obtained in their first interaction with the robot, which was performed in a similar way as the first phase: after giving them brief instructions, they interacted freely with the robot for ten minutes while being observed by a therapist. We expect to perform another round of tests with the same elderly users in the following months to compare the evolution of the user’s acceptance after long-term interaction with the robot. This round was expected to be performed earlier but the experimentation had to be halted due to the COVID-19 pandemic.

### 3.2. Results Obtained

The results obtained in the questionnaires from the adult users’ tests are reported in Table 1, which shows the scores obtained in each evaluated aspect. From the results, it is inferred that the participants were comfortable with the robot, as reflected by the low anxiety value obtained, and that it transmitted confidence to them. It is also observed that the participants, in general, had a positive attitude towards the robot, found its use simple and found it useful and entertaining. On the other hand, the results show that the participants perceived that the robot does not have “real intelligence”, feeling that sometimes the robot does not understand what they want to say or that it is not able to hold a meaningful conversation.

The results obtained from the elder care facility users’ tests are shown in Table 2. The most notable differences compared to the previous results, as can be seen on Figure 8 and Figure 9, are that elderly users show a higher anxiety and lower facilitating conditions and also felt the robot less easy to use. This can be explained by the digital divide, as elderly people are less used to the use of technology than younger people. Although the previous experience with technical devices was not assessed for the participants, given the participants’ age range (minimum 76, maximum 99, average 88), it is safe to assume they had little experience.

The elderly users showed higher perceived usefulness and slightly higher social presence than adults. The other parameters do not show significant differences between the two groups.

Overall, the elderly users showed a positive attitude towards the robot and enjoyed the time spent with it.

The results obtained regarding the evaluation of the different robot features are shown in Table 3.

As can be seen on Figure 10, overall, both the adult and elderly users made a positive evaluation of the robot features. Music and Exercises were more appreciated by the elderly users, which can be explained because they are specifically designed for the elderly and younger users might not feel motivated by them. The orientation games are also more valued by the elderly users, mainly because of the balloons game, which is very appreciated by them. Voice recognition is also more valued by the elderly users, as younger users can more easily notice the limitations of the system. On the other hand, the language games were more appreciated by the adult users because, initially, only the “riddles” game was present in this category, which elderly users could find more difficult to understand. In order to overcome this drawback, we added the “sayings” game, as suggested by the therapists, as it is more appropriated for them.

## 4. Discussion

In recent years, remarkable progress has been made in research on service robots, and particularly on social robots. A social robot interacts and communicates with people, in a simple and pleasant way, following behaviors, patterns and social norms. There has also been a growing interest in integrating techniques from the areas of artificial intelligence, computer vision, navigation, manipulation, learning, planning, natural language, voice recognition and empathic interaction. One of the most attractive ideas is that robots should not only be taken as tools, but also as companions to humans. Social robots have to (i) interact with humans and their environment in a friendly way, just as people do, and (ii) offer useful services at a reasonable cost.

The obtained results have shown a positive acceptance of the solution both on adult and elderly users with many similarities between them. The main differences found are in the anxiety and facilitating conditions aspects that, as mentioned before, can be explained by the digital divide. As elderly users will become more used to the use of modern technologies and social robots, this difference will disappear, and the acceptance of the robot solution will become even more positive. We have observed that when users understand the functionalities and the way to communicate with the robot, they become more confident. In those cases, the interaction could be developed without the intervention of a caregiver. We hope that in the next studies of a user’s acceptability, when they become more experienced, the anxiety could have lower levels and the facilitating conditions result could have improved.

The use of social robotics, according to the European Union, is part of the solution for the future of caring for the elderly. However, robots interacting with people for long periods of time are very rare. This is for the following two main reasons: i. there are still perceptual and cognitive limitations of robots that hinder their autonomous functioning and ii. most of the research is being carried out in laboratory settings and not in real human operating environments, where the needs and difficulties of interaction can be appreciated. Although, in our opinion, it is necessary, avoiding any risk, to advance in experimentation in real scenarios, as in geriatrics residences, in which the gerontologist could detect any risky situation during the interaction.

Other studies have also addressed the usability of a social robots for elder care. Cognitive therapies using robots were addressed in [17], obtaining promising results as the users were becoming more motivated on each session during the period of 6 months in which the tests were performed. In [18] and [28], they achieved positive results with robot exercise coaches, although the interactions were conducted during short time periods. On the other hand, in [29], they found that the robot was not appealing to the users, and they were not motivated to use it, remarking the relevance of the digital divide for the acceptance of robots by elderly users. In addition, long-term interaction studies have found that although the robot can be appealing in the short-term, it is important that the robot has relevant functions to ensure that the users keep motivated and continue using it in the longer-term [30,31]. In our project, we have developed a robotic platform that addresses both physical and cognitive stimulation and, although in the present paper we are presenting the initial usability validation with positive results, we will be obtaining interaction data of a long-term interaction period of over a year in the following months. Even though the experimentation had to be halted due to the COVID-19 pandemic, one robot is being kept permanently in the elder care facility, interacting with the users and being continuously improved through remote updates based on the feedback of the interaction with the users received from the therapists, ensuring that the robot is useful, and users keep motivated. Some drawbacks have been addressed thus far, such as the difficulty of some users to understand the robot’s voice, which we mitigated by adding the possibility of customizing the robot voice’s pitch and speed for each user. We also added an option to indicate if the user is being helped by the caregiver while performing a therapy, allowing us to obtain more accurate results when following the user’s evolution.

Concerning the progress beyond the state of the art, this project is expected to generate a solution to help in the integration of social robots at home in the near future, the development of services in the field of assistance to dependent people, human machine interaction and robotics in the cloud. In particular, this progress will come hand in hand with the automatic learning of social behaviors and the adaptation to the user by combining the latest techniques of machine learning (including deep learning), quantifying the behavior of social robots through metrics that can be used for the continuous improvement of interaction and services, and the use of the cloud not only as a computing and storage resource, but also as a place for the exchange of experiences and knowledge among robots.

## 5. Conclusions

Our preliminary results are satisfactory, not only in terms of acceptance by users, as seen in the results, but also, the feedback we are receiving from the therapists shows that the platform can be an effective solution to keep them physically and mentally active. In this sense, the current crisis of the socio-health system has had a strong impact in gerontological centers due to the Coronavirus SARS-CoV-2 pandemic. The restriction measures that CoV-2 has entailed, such as the limitations on visits and interaction between residents caused by social distancing, which has restricted group activities in many cases, has led to very significant detrimental psychologic effects. Lacort’s gerontologists have been able to partially alleviate this extreme situation by using the robot, which they nicknamed “Copito” (a name chosen by the users of the center because it is white as a snowflake).

During the following months, we will be performing another Almere test within the elder care facility users and studying the evolution of the users’ acceptance of the solution. In addition, we will study the evolution of the results obtained in the MMS tests to see if long-term interaction with the robot has an impact on them (by comparing with other users that have not used the robot) and if the evolution in the MMS scores can be correlated with the results gathered from the interaction with the robot (for instance the scores obtained in the different game categories over the time).

## Figures and Tables

**Figure 1 healthcare-09-01067-f001:**
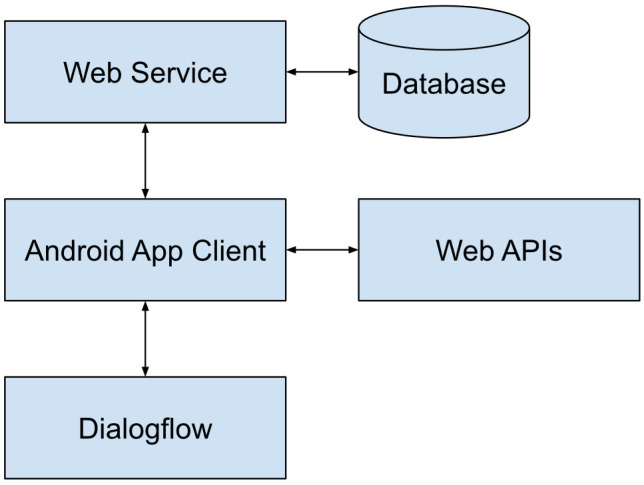
System architecture overview.

**Figure 2 healthcare-09-01067-f002:**
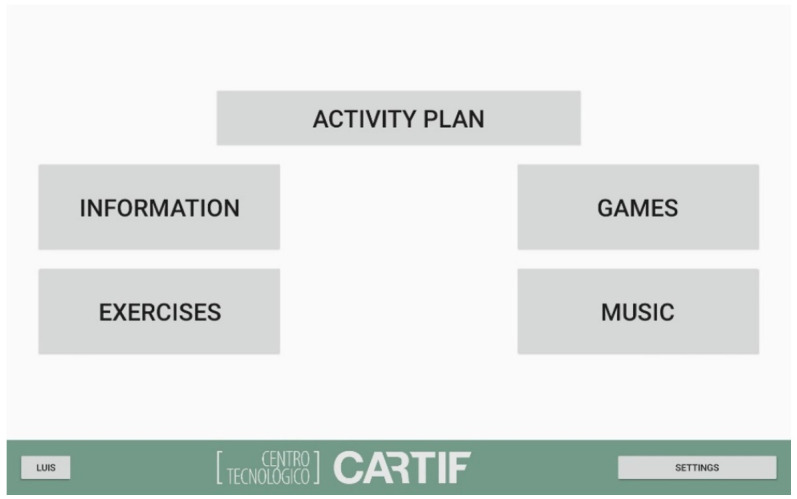
Main menu.

**Figure 3 healthcare-09-01067-f003:**
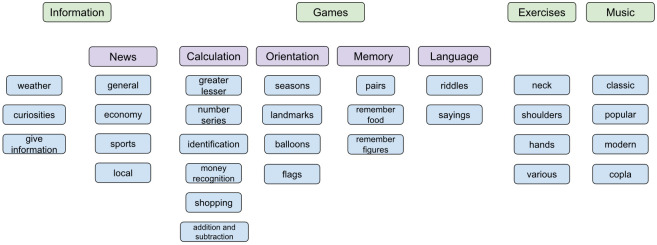
Features overview.

**Figure 4 healthcare-09-01067-f004:**
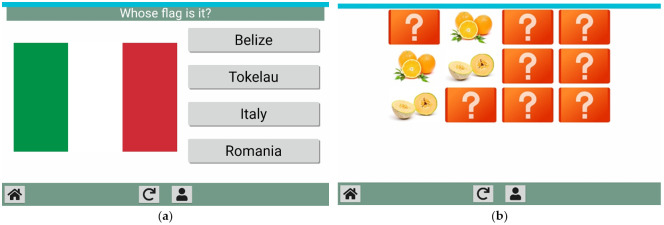
Examples of the “flags” orientation game (**a**) and the memory game “pairs” (**b**).

**Figure 5 healthcare-09-01067-f005:**
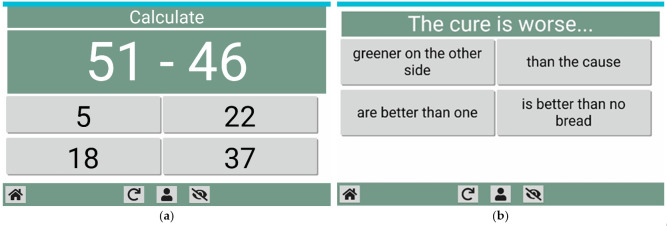
Examples of the “addition and subtraction” calculation game (**a**) and the “sayings” language game (**b**).

**Figure 6 healthcare-09-01067-f006:**
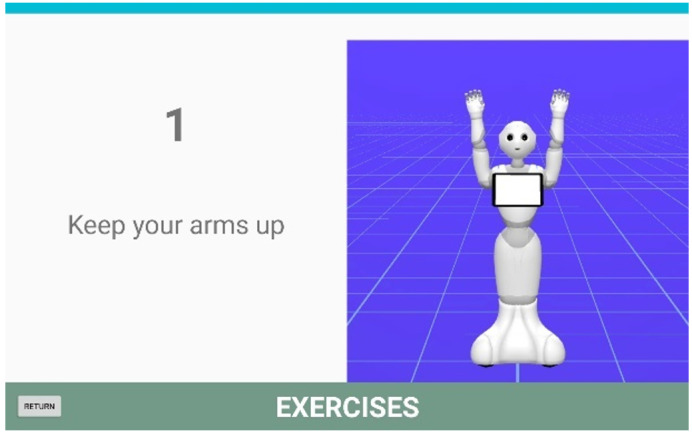
Example of shoulder exercises.

**Figure 7 healthcare-09-01067-f007:**
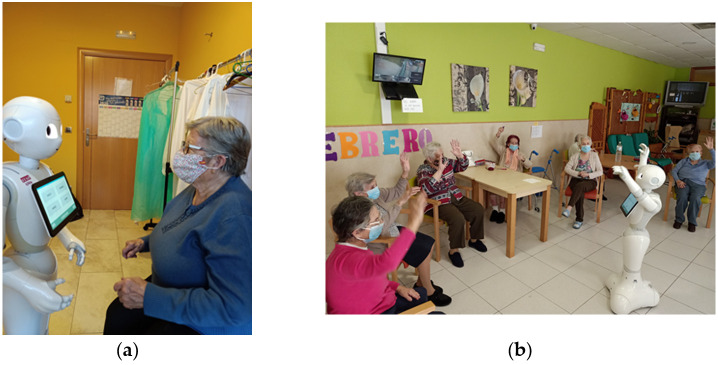
Users interacting with the robot in Lacort elder care facility individually (**a**) and performing exercises in group (**b**).

**Figure 8 healthcare-09-01067-f008:**
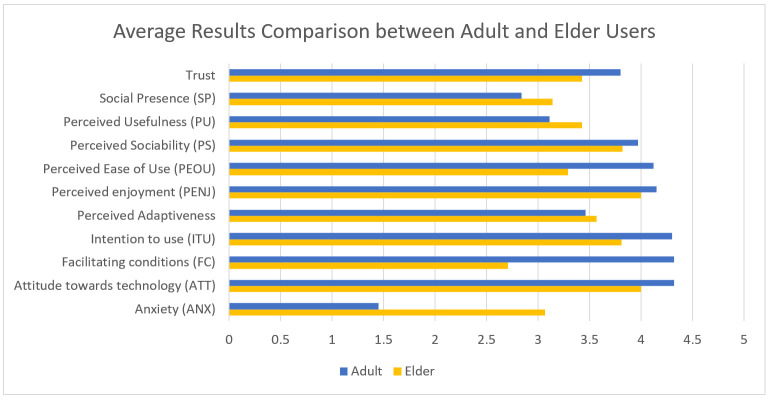
Average Results Comparison between Adult and Elderly Users.

**Figure 9 healthcare-09-01067-f009:**
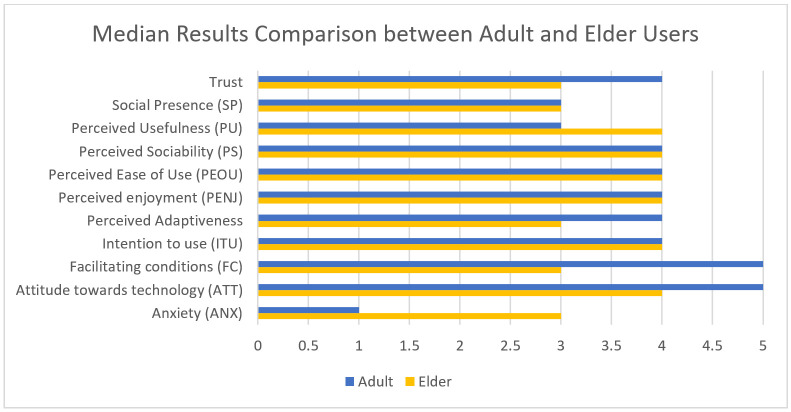
Median Results Comparison between Adult and Elderly Users.

**Figure 10 healthcare-09-01067-f010:**
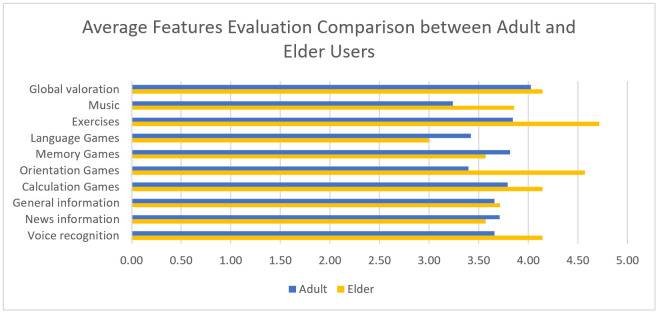
Average Features Evaluation Comparison between Adult and Elderly Users.

**Table 1 healthcare-09-01067-t001:** Results of the questionnaires from the adult users’ tests.

Aspect	Average	Deviation	Median
Anxiety (ANX)	1.45	0.90	1
Attitude towards technology (ATT)	4.32	0.93	5
Facilitating conditions (FC)	4.32	0.87	5
Intention to use (ITU)	4.30	1.12	4
Perceived Adaptiveness	3.46	1.01	4
Perceived enjoyment (PENJ)	4.15	0.90	4
Perceived Ease of Use (PEOU)	4.12	1.11	4
Perceived Sociability (PS)	3.97	1.05	4
Perceived Usefulness (PU)	3.11	1.22	3
Social Presence (SP)	2.84	1.31	3
Trust	3.80	1.08	4

**Table 2 healthcare-09-01067-t002:** Results of the questionnaires from the elder care facility users’ tests.

Aspect	Average	Deviation	Median
Anxiety (ANX)	3.07	1.12	3
Attitude towards technology (ATT)	4.00	1.15	4
Facilitating conditions (FC)	2.71	0.76	3
Intention to use (ITU)	3.81	1.07	4
Perceived Adaptiveness	3.57	0.68	3
Perceived enjoyment (PENJ)	4.00	1.00	4
Perceived Ease of Use (PEOU)	3.29	1.23	4
Perceived Sociability (PS)	3.82	0.86	4
Perceived Usefulness (PU)	3.43	0.94	4
Social Presence (SP)	3.14	1.11	3
Trust	3.43	0.94	3

**Table 3 healthcare-09-01067-t003:** Results of the robot features evaluation.

	Adults		Elders	
Feature	Average	Deviation	Average	Deviation
Voice recognition	3.66	1.05	4.14	0.90
News information	3.71	1.35	3.57	0.98
General information	3.66	1.26	3.71	1.25
Calculation games	3.49	1.23	4.14	0.90
Orientation games	3.39	1.67	4.57	0.79
Memory games	3.82	1.35	3.57	1.13
Language games	3.42	1.67	3.00	1.15
Exercises	3.84	0.89	4.71	0.49
Music	3.24	1.22	3.86	0.90
Global evaluation	4.03	0.64	4.14	0.69

## Data Availability

Not applicable.

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
