# Peer review of "Development and Usability Validation of a Social Robot Platform for Physical and Cognitive Stimulation in Elder Care Facilities"

_healthcare, 2021, doi:10.3390/healthcare9081067_

Round 1
Reviewer 1 Report
Title: Development and usability validation of a social robot platform for physical and cognitive stimulation in elder care facilities
Contribution: elder care platform for social interaction and physical/cognitive stimulation using the Pepper robot and Adroid OS client. Interactive (speech and touch screen) services provided. Adults and elder test and validation. Data stored on a cloud database. Proposal of Mini-Mental State examination MMS as acceptable scored Mental State Examination MSE for elderlies, based on speech and written interaction.
Overall: Readable and clear document. State of the art considering other related studies is missing. The validation doesn’t include duration, activities, type of interaction (voice/touch screen), type of information stored, language used (and language difficulties).
Author Response
Dear reviewer,
Thanks for the review. We appreciate your suggestions, allowing us to improve the paper.
Regarding the state of the art considering related studies, we have added more references of similar works from other researchers in the introduction and the discussion.
Regarding the validation, we have added to the results section that in both phases the users interacted with the robot in their native language (Spanish). We have also added data from the participants of the elder care facility: number of participants, average age and average minimental score. We also added more details from the experiments and the procedure for their first interaction with the robot, which was similar to the first phase.
Regards,
The research group
Reviewer 2 Report
Comments on the content:
The article reports user trials with a social robotic platform for social interaction and physical and cognitive stimulation of elderly. This is a timely and relevant topic of research.
I would include in the introduction the rationale for choosing a humanoid robot instead of installing in the environment several tablets/displays or interfaces for voice assistants (different studies can be cited showing that, in general, people interact more with a humanoid robot than with a tablet). The introduction should also briefly review the literature on the use of social robots for elderly care. At a minimum, it should refer to similar studies that have been conducted by other research groups.
On line 60 you have “The main target of this work is to develop and validate a social robot platform using Pepper robot and Android OS as clients that allows it to serve as an assistant robot for elder care, making it able to interact socially with them, entertain them and keep them physically and mentally active.” I believe this describes the end goal of your project but not the scope of the submitted article. The study presented in the article is a first step towards establishing user acceptance. In my opinion, that should be made clear (as it is in the Abstract).
In my opinion, you should prepare the reader for section 2.2. on the Mini-Mental State Examination. Up to that section, Peper was presented as “an elder care platform for social interaction and physical and cognitive stimulation” (from the abstract). The description of MMS looks out of context.
The rationale for choosing the features of the system should be explained: why was it decided that the robot would provide information, play games, mediate physical exercises, and play music? Was it based on requirements of potential users? Of therapists? Are these just a first set of features to prove the concept and then further studies will be conducted to understand what are the desirable features of the robot? For example, the robot could read a book… In my opinion, it’s absolutely critical to involve the end-users in all stages of the design and not only when it comes to trial the system developed based on what others believe the end-users need and want.
Section 3.2.: Before interpreting the results, it would be useful for the reader not familiar with the Almere model, to state that questions were answered using a 5-points Likert scale from 1 to 5 (totally disagree – disagree – don’t know – agree – totally agree) (I’m guessing this was the case). Only knowing that, one can understand the interpretation of the results.
The participants in the study are poorly described. Which was the age range? How many persons interacted with the robot in the Lacort Viana elderly care facility? Did any of the participants had any limitation that may have impacted the interaction with the robot? Apparently, that was the case, since you wrote on line 292 “highlighting the difficulty of working with elderly people with perceptive and, in some cases, cognitive limitations”. In my opinion, a detailed description of the participants needs to be included so the reader can assess the significance of the results.
As far as I understand (that needs to be clarified in the paper!), each participant interacted only once with the robot. This contrasts with the envisioned use of such robotic tools in which the robots are seen as social companions. The discussion should address this limitation of the presented study. Lines 276-284 discuss the need for longitudinal studies and authors state that, in their opinion, such longitudinal studies should be conducted (lines 281-284), but then it is not clearly stated that the study reports the results of a single interaction of each participant with the robotic platform. But perhaps the robot stayed for a longer period in the centre since you mention that therapists are using the robot to ameliorate the detrimental psychologic effects of the COVID-19 pandemic… Please clarify.
It is hypothesized that the different results obtained with younger adults and elderly may be due to digital divide. Do you have data that can substantiate that hypothesis? For example, have you registered the participants’ experience with technology devices? If so, that information should be shared.
The discussion should compare the results obtained with the results obtained in similar studies by other research groups.
In the conclusions (line 305), you state “but indicators are being observed which show that the platform it as an effective solution to keep them physically and mentally active”. This was not discussed in the paper. As mentioned before, if there was a single interaction between each participant and the robot, which can be those indicators? Please either discuss those “indicators” in the paper or consider removing the sentence from the conclusions.
Also in the conclusions (line 310), you state that COVID-19-related constraints “has led to a very significant cognitive impairment.” I believe you should be careful with the words. Perhaps substitute “cognitive impairment” by “detrimental psychologic effects”.
The last sentence of the conclusions (lines 315-317) states that you plan to “correlate the data obtained from the minimental test and the results of the interaction with Copito.” It is not clear to me what you mean. Do you intend to investigate if the use of the robot has an impact on the results of the MMS? Please clarify.
If the study was approved by an Ethics Committee, please include that information. If not, please include a sentence explaining why an Ethics Committee approval could be waived.
Comments on the form:
Line 74: Please define the acronym SDK before using it (“Limited SDK” -> “Limited Software Development Kit (SDK)”)
Line 87: Should it be “2.2. Mini-Mental”? Perhaps it would be clearer to have “2.2. Mini-Mental State examination”
Line 99: “in [12] recommends” -> “in [12] recommend”. Please check the rest of the sentence (“…(MSE) in this they recommend…”?)
Line 105: Please check the sentence. Would it be better to write “The questions in the MMS questionnaire should be asked in…”?
Line 114: Please check the sentence (“…and allowed for in the scoring.”?)
Line 161: Please include a reference for the JSON format.
Line 173: “using the speech and/or by the touch screen interface” -> “using speech and/or the touch screen interface”
Line 184: I suggest rewording the sentence to “The system also allows the therapists to set up customized activity plans for each user or group of users through a web interface.”
Line 207: I would remove “with real people”. The participants are described in the following sentence.
Line 209: I would remove the word “real”.
Line 216: I would substitute “being surveilled externally during the interaction” by “while being videotaped / while being observed”, whichever was the case.
Line 230: Please reword the sentence. I failed to understand what you mean by “grouped according to their corresponding aspect”.
Line 238: I would prefer “to hold a meaningful conversation” instead of “real conversation”. But you should keep as it is if that was the question asked to the users.
Line 290: “a set metrics” –> “a set of metrics”
Line 291: I believe it is not clear what you mean by “common users” and “users”. Are you referring to the comparison between “adults” and “elderly” performed? If so, please use the same terminology.
Line 310: “which has restricted in many cases the perform of group activities” -> “which has restricted in many cases group activities”
Line 317: Should it be “mini-mental” or “MMS”?
References: Please change the language to English when automatically creating the list of references (e.g., you have “En…” instead of “In…”). I would capitalize each word in the names of the journals and proceedings of conferences.
Reviewer 3 Report
The paper purport to determine usability validation of a social robot. The topic is very relevant, however the paper reports extensively the technical and information providing part, but its description of the experimentation is very limited.
Two groups of participant were involved: 1 staff from a technology centre and 2 residents of a care facility. The experiment for group 1 lasted 10 minutes; but no numbers have been given for group 2x. For group 2, an overview of experimentation time should have been given, which is their native language?. With this information the reader would be able to judge the statement that they get more confident (line 272).
Also it is not clear whether group 2 were presented with English or Spanish interfaces (shown in figure 5); this matters considerably when comparing tables 1 and 2.
The discussion section is quite long and it is not clear why for instance lines 285-293 appear in this section.
The English language is understandable, but clearly not from a native speaker. For instance makes necessary the improvement (l 38-39) and a similar statement in line 42.
Author Response
Dear reviewer,
Thanks for the review. We appreciate your suggestions, allowing us to improve the paper.
Regarding the experimentation, we have added to the results section that in both phases the users interacted with the robot in their native language (Spanish). We have also added data from the participants of the elder care facility: number of participants (21), average age (88) and average minimental score (23). We also added more details from the experiments and the procedure for their first interaction with the robot, which was similar to the first phase.
Regarding the discussion, we have removed some lines as suggested and added references to other similar works.
Regarding the English language, we have checked and fixed the statements you mention and some others found throughout the document.
Regards,
The research group
Round 2
Reviewer 1 Report
The results are still almost the same, it should be extended according to the different cathegories and usages to clearly identify drawbacks, improvements, and (positive/negative/subtile) evolution of patients.
Author Response
Dear reviewer,
thank you for your comments and suggestions. Our line of research aims to develop and validate a social robot platform using Pepper robot and Android OS as clients that allows it to serve as an assistant robot for elder care, making it able to interact socially with the users, entertain them and keep them physically and mentally active. In this article we are presenting the first results of the user's acceptance of the solution, which we have evaluated using the almere model.
In the following months we will be getting long-term interaction results, performing another Almere test within the elder care facility users and studying the evolution of the users’ acceptance of the solution. In addition, we will study the evolution of the results obtained in the MMS tests to see if long-term interaction with the robot has an impact on them (by comparing with other users that haven’t used the robot) and if the evolution in the MMS scores can be correlated with the results gathered from the interaction with the robot (for instance the scores obtained in the different game categories over the time). Those long-interaction results were expected to be gathered earlier but the experimentation had to be halted due to the COVID-19 pandemic.
We have clarified that this article addresses the first results of the user's acceptance of the solution in the begining of the matherials and methods section (lines 69-73). It has been also clarified on the results section (lines 261 to 267), discussion (lines 363 to 375) and conclussion (lines 408 to 416). Although we will be getting long-term results in the following months, as one robot is kept on the elder care facility interacting with the users we have appointed in the discussion somefeedback that we are getting from the users and caregivers, as well as drawbacks that have been addresed and improvements made based on that feedback.
We have also extended the results section with an evaluation of the different robot features made by the adult and elder users (lines 305 to 321). They fulfilled that evaluation together with the almere questionnaire in order to get feedback about which of the features were the most appreciated and identify drawbacks. We initially planned to include only the almere results, but we have now included those as you requested in order to showcase how the users evaluated the robot features and the drawbacks we found.
Regards,
The research team
Reviewer 2 Report
I would like to thank the authors for editing the article according to the reviewers’ suggestions.
It is my opinion that the paper can now be published, upon minor editing (see suggestions below).
Editing suggestions:
Line 30: “that affect to their ability” -> “that affect their ability”
Line 144: “The view part of the system” -> “The interface to the system”
Line 176: “show” -> “shows”
Line 187: Please include here the reference [26] to the JSON format, instead of on line 191
Line 239: If you have the average age of these participants, please include it here so one can compare it with the one from the other group
Line 263: “surveilled” -> “observed”
Line 286: Still the issue on the digital divide… Perhaps you could add here something along these lines: “Although the previous experience with technical devices was not assessed for the participants, given the participants age range (minimum x, maximum y, average 88), it is safe to assume they had little experience.”
Author Response
Dear reviewer,
thank you for your editing suggestions, we have corrected all of them as you have proposed.
By request of reviewer 1 we have also extended the results section with an evaluation of the different robot features made by the adult and elder users (lines 305 to 321). They fulfilled that evaluation together with the almere questionnaire in order to get feedback about which of the features were the most appreciated and identify drawbacks. We initially planned to include only the almere results, but we have now included those as reviewer 2 has requested in order to showcase how the users evaluated the robot features and the drawbacks we found.
Regards,
The research team